# Molecular detection and risk factors of *Eimeria* in native and exotic chickens under varying management systems in Bangladesh

**Md. Afazur Rahman[1], A R M Beni Amin[1], Kazi Farah Tasfia[1], Makoto Matsubayashi[2], Md. Shahiduzzaman●[1]***

**1** Department of Parasitology, Faculty of Veterinary Science, Bangladesh Agricultural University, Mymensingh, Bangladesh, **2** Graduate School of Veterinary Medical Sciences, Osaka Metropolitan University, Osaka, Japan

* szaman@bau.edu.bd

## Abstract

A cross-sectional study was conducted in Bangladesh to determine the prevalence, molecular detection, and risk factors of *Eimeria* spp. infection in native and exotic chickens under various management systems. A total of 1,500 fecal samples were collected from different breeds, age groups, and sexes across multiple districts. Fecal examination using flotation and McMaster techniques identified positive cases, followed by molecular detection of *Eimeria* species. A questionnaire survey was also conducted to assess potential risk factors. Among the 1,500 chickens, 87 (5.8%) were positive for *Eimeria* oocysts, with higher prevalence in exotic breeds (7.96%) than native breeds (4.13%). The prevalence rates were 18.40%, 13.98%, 12.09%, and 3.40% in Aseel, Broiler, Sonali, and Deshi chickens, respectively, with no infection found in Naked Neck, Hilly, or Fayoumi breeds. Molecular analysis detected six *Eimeria* species: *E. tenella* was detected in 64 samples (62.07%) and in all breeds with the highest occurrence in Aseel. *E. acervulina* was the second most common species (25.28%), found in 23 samples from Deshi, Broiler and Sonali breeds. Other species, including *E. brunetti*, *E. mitis*, *E. necatrix*, and *E. maxima*, were rare and sparsely distributed. Chickens fed commercial feed (7.88%) had significantly higher infection rates (p < 0.0013) than those on local feed (3.99%). Intensive rearing systems (15.27%) showed higher infection rates compared to free-ranging systems, but no infection occurred in intensive systems without litter or semi-intensive systems. This is the first comprehensive report on infection status of *Eimeria* in chickens including all native breeds rearing in different management system in Bangladesh.

## Introduction

The poultry industry in Bangladesh has experienced rapid growth over the past few decades, transitioning from traditional backyard farming to more intensive and

**Data availability statement:** All relevant data are within the manuscript.

**Funding:** This study was funded by the Bangladesh Academy of Sciences under the BAS-USDA Program (BAS-USDA Program/2023/26(12) to M.S.). The funder had no role in study design, data collection and analysis, decision to publish, or preparation of the manuscript.

**Competing interests:** The authors have declared that no competing interests exist.

commercialized systems. This expansion has significantly contributed to economic growth, job creation, and the availability of affordable protein sources [1]. According to the Bangladesh Poultry Industries Central Council (BPICC), the country produces approximately 1.5 million metric tons of chicken meat and over 10 billion eggs annually [2]. However, this growth has been accompanied by an increased prevalence of poultry diseases, particularly coccidiosis, which poses a significant challenge to both commercial and backyard farming systems.

Coccidiosis, caused by protozoan parasites of the genus *Eimeria*, is one of the major diseases affecting poultry worldwide. The warm and humid climate of Bangladesh, combined with suboptimal management practices and inadequate biosecurity measures, creates favorable conditions for *Eimeria* proliferation. The disease leads to considerable economic losses due to increased mortality, reduced body weight, and expenses associated with preventive and therapeutic control measures [3].

Several factors contribute to the high prevalence of coccidiosis in Bangladesh, including poor housing conditions, overcrowding, improper litter management, and unregulated use of anticoccidial drugs. Additionally, non-scientific rearing practices, such as mixing old and new litter and maintaining short rest periods between flocks (8–14 days), further increase disease transmission risks [4,5]. Poor biosecurity practices, such as farm visitors, live bird markets, and off-site worker accommodations, also elevate infection risks [6].

Poultry farming in Bangladesh operates under two primary systems: family-based (traditional backyard) farming and commercial production (small, medium, and large-scale farms) [7,8]. Backyard poultry farming primarily involves indigenous breeds such as Naked Neck, Deshi, Aseel, and Hilly chickens, which play a crucial role in rural livelihoods by providing food security and supplemental income. These low-input systems are integrated into household activities, relying on scavenging with minimal external feed supplementation. In contrast, commercial farms focus on high-yielding breeds such as Cobb 500 broilers and specialized layers. Sonali chickens, a cross-breed of Rhode Island Red and Fayoumi, are particularly popular for their adaptability and superior meat quality [9]. However, small and medium-sized farms often face challenges such as dependence on middlemen, inadequate disease management knowledge, and misuse of antibiotics and anticoccidials [10,11].

Accurate identification of *Eimeria* species is crucial for effective disease management. Traditional microscopic methods often fail to distinguish between different species, leading to diagnostic inaccuracies. Molecular techniques, such as PCR-based identification, provide higher sensitivity and specificity, enabling precise characterization of *Eimeria* species affecting poultry [12–14].

While previous studies have investigated coccidiosis in commercial broilers and Sonali chickens [5,9,15–17], there is limited data on indigenous breeds. Additionally, existing research has primarily focused on specific regions, leaving gaps in understanding the disease's prevalence across diverse agro-ecological zones of Bangladesh. The influence of different rearing systems (backyard, semi-intensive, and commercial) on coccidiosis prevalence and severity remains underexplored. A

comprehensive risk analysis across various chicken breeds and management practices is essential for sustainable and cost-effective poultry production.

This study aims to provide a comprehensive assessment of *Eimeria* infections in native and exotic chickens under different management systems in Bangladesh. The specific objectives are:

1. To determine the prevalence of *Eimeria* infections in different breeds of chickens across various poultry farming systems.

2. To molecularly characterize *Eimeria* species using PCR-based techniques for accurate identification.

3. To analyze the key risk factors associated with coccidiosis prevalence, including feeds, rearing systems, and breeds.

The findings of this study will provide valuable insights for poultry farmers, veterinarians, and policymakers to enhance disease management practices. By identifying high-risk farming systems and chicken breeds, the study will contribute to the development of targeted interventions, ultimately improving the productivity and sustainability of the poultry industry in Bangladesh.

## Materials and methods

### Description of study area and samples

This study was conducted across six major divisions of Bangladesh: Mymensingh, Rangpur, Dhaka, Shariatpur, Comilla, Rajshahi, Bogura, Narsingdi, and Chattogram (Table 1). A total of 1,500 fecal samples were collected from these regions to assess the prevalence of chicken coccidiosis and the associated risk factors. This comprehensive geographical coverage ensures a representative understanding of the prevalence of coccidiosis across different poultry farming systems in Bangladesh. Samples were collected from both exotic breeds (Sonali, Broiler, Fayoumi) and native breeds (Deshi, Aseel, Naked Neck and Hilly) of chicken.

### Current anticoccidial practices in Bangladesh

Commercial farms raising both exotic and native chickens commonly use anticoccidial feed purchased from various companies, except for chickens reared in household free-range systems. These commercial feeds typically include anticoccidials, antibiotics, and growth promoters. The current anticoccidial practices in Bangladesh involve a rotational shuttle program combining ionophore and non-ionophore chemicals. The strategy includes the use of combinations such as maduramycin with nicarbazin, salinomycin with nicarbazin, monensin with nicarbizin, and sanduramycin with nicarbizin in alternating months (3 months each combination). Additionally, Decoquinate is used for one month, followed by Ethopabate for one month, and Clopidol for another month, completing a year-long shuttle program to manage coccidiosis effectively (Personal communication with company and farmers).

### Fecal sample collection and oocysts observation

A total of 1,500 fecal samples were collected from 77 farms or flocks across nine districts in six major divisions of Bangladesh. From each farm or household, approximately 15–25 fecal samples were collected depending on flock size. Fresh fecal samples (approximately 7–10 grams) were collected by using individual wood stick from freshly voided feces randomly from the flock. Each fecal sample was placed in a plastic container and marked. To maintain diagnostic resolution, each sample was kept and processed individually; no pooling was performed. The age (young and adult), breed, sex, rearing system, and feed were noted against the corresponding marking of the samples (Table 2). Samples from male or female bird were collected upon identification of sex. The samples were then transported through icebox to the laboratory at the Department of Parasitology, Bangladesh Agricultural University for faecal examination. Samples were stored in the refrigerator at 4°C if not immediately examined.

**Table 1. The specific locations within each region and their respective geographical coordinates of sample collection areas.**

| Locality/region | Longitude and latitude | No of farms/flocks (Total chicken) | Samples (N)/ examined birds |
|---|---|---|---|
| Mymensingh | Sadar (Town): 24°43'42.08"N, 90°24'28.86"E<br>Shomvugang: 24°45'44.71"N, 90°26'53.12"E<br>Vabokhali: 24°40'36.25"N, 90°26'50.54"E<br>Trishal: 24°35'6.15"N, 90°23'17.20"E<br>Chor: 24°43'52.31"N, 90°27'25.10"E<br>Sutiakhali: 24°41'29.74"N, 90°26'51.65"E<br>Kalir Bazar: 24°37'55.84"N, 90°27'50.22"E<br>Nalitabari: 25° 5'5.23"N, 90°13'11.68"E | 17 (6500) | 478 |
| Dhaka | BLRI (Savar): 23°53'19.31"N, 90°16'26.58"E<br>Savar: 23°53'52.40"N, 90°15'49.36"E<br>Rajeer: 23°12'23.27"N, 90° 3'7.40"E<br>Madaripur: 23° 9'44.83"N, 90°12'12.27"E | 10 (2270) | 115 |
| Comilla | Sarail: 24° 4'19.23"N, 91° 6'55.02"E<br>Muradnagar: 23°39'34.83"N, 90°54'23.81"E<br>Debidwar: 23°36'43.73"N, 90°58'0.31"E<br>Burichang: 23°33'7.59"N, 91° 7'10.08"E<br>Laksham: 22°57'5.12"N, 90°52'4.03"E<br>Titas: 23°35'45.08"N, 90°50'14.18"E | 9 (2160) | 267 |
| Shariotpur | Sadar (Town): 23°23'94N, 90°37'07E<br>Amtoli: 23°18'72N, 90°35'82E | 4 (1500) | 95 |
| Rangpur | Gangachara: 25°50'27.12"N, 89°13'45.70"E<br>Haragach: 25°49'11.15"N, 89°19'46.66"E<br>Mithapukur: 25°34'38.39"N, 89°16'50.97"E | 6 (1650) | 155 |
| Rajshahi | Puba: 24°25'26.40"N, 88°39'26.99"E<br>Bhangura: 24°12'28.98"N, 89°21'57.59"E | 4 (250) | 85 |
| Bogura | Shahbandegi: 24°38'49.74"N, 89°25'9.97"E<br>Town Kaleni: 24°50'0.22"N, 89°22'25.14"E | 8 (730) | 95 |
| Narshingdi | Birpur: 23°55'44.48"N, 90°44'7.76"E<br>Shivpur: 24° 1'43.78"N, 90°43'43.95"E | 6 (240) | 72 |
| Chattogram | Alikadam Bazar: 21°38'44.37"N, 92°18'26.68"E<br>Matiranga Bazar: 23° 2'33.43"N, 91°52'19.97"E<br>Marma Borokholo: 22°23'31.92"N, 92° 3'12.85"E | 13 (860) | 138 |
| **Total** | | **77 (16160)** | **1500** |

Note: At least 10 birds were examined from each farm/flock

At the day of examination, the sample of an individual chicken was blended by a mortar and pistol, and then the floatation technique [18] was applied using saturated sodium chloride solution to concentrate oocysts aiming oocyst screening (oocysts per gram-OPG) and downward application. Five grams of homogenized droppings were mixed with 45 ml NaCl saturated salt solution (density = 1.20 g/ml) and filtered to remove coarse particles. Processed solution was poured through a tea strainer into a beaker then into a 15 ml centrifuge tube. For quantification, the McMaster counting technique as outlined by Hodgson [18] was used. A 0.3 ml aliquot of the fecal suspension was loaded into a McMaster counting chamber, filling both grids (0.15 ml per grid). Later the centrifuge tube was covered with a cover slip and allowed to stand for 20 minutes and then removed and placed on a slide and examined at 10x and then 40x magnifications to identify the oocyst [19].

## Molecular detection of *Eimeria* species

To determine the species of *Eimeria* prevailing in different chicken farms of Bangladesh, the microscopically positive samples were subjected to DNA extraction and PCR. DNA was extracted using DNAzol Reagent (Invitrogen, USA) according

**Table 2. Prevalence of *Eimeria* infection based on chicken-related risk factor.**

| Risk factors | Categories | No of samples examined | No. of positive samples (%) | Average OPG (range) | No of farms/flocks (coccidiostat used) | No of positive farms/flock (%) | Significance (p<0.05) |
|---|---|---|---|---|---|---|---|
| Exotic Breed | Sonali | 182 | 22 (12.09) | 133–1066 | 12(12) | 8 (66.67) | * |
| | Broiler | 186 | 26 (13.98) | 33–1967 | 13(13) | 10 (76.92) | * |
| | Fayoumi | 235 | 0 | 0 | 5(3) | 0 | |
| | Subtotal | 603 | 48 (7.96) | | | | |
| Native Breed | Deshi | 382 | 13 (3.40) | 33–133 | 18(1) | 7 (38.89) | |
| | Aseel | 125 | 23 (18.40) | 66-500 | 5(5) | 5 (100) | * |
| | Naked Neck | 227 | 0 | | 15 (1) | 0 | |
| | Hilly | 138 | 0 | | 5 (0) | 0 | |
| | Subtotal | 872 | 36 (4.13) | | | | |
| Mixed population | Mixed (Sonali+Deshi) | 15 | 1 (Sonali, 6.67) | 66–100 | 2(1) | 1 (50.00) | |
| | Mixed (broiler+Deshi) | 10 | 2 (Broiler, 20.0) | 33–133 | 2(2) | 1 (50.00) | * |
| | Subtotal | 25 | 3 (12.0) | | | | |
| Sex | Male | 255 | 25 (9.80) | | | | * |
| | Female | 1245 | 62 (4.98) | | | | |
| Age | Adult (>60 days) | 334 | 2 (0.17) | | | | |
| | Young (1–60 days) | 1166 | 85 (25.45) | | | | * |
| Rearing system | Intensive with litter | 478 | 73 (15.27) | | 25 (20) | 17 (68) | * |
| | Intensive without litter | 172 | 0 | | 7 (5) | 3 (42.86) | |
| | Semi-intensive with litter | 50 | 0 | | 10 (8) | 3 (30.00) | |
| | Semi-intensive without litter | 154 | 0 | | 5 (3) | 2 (40.00) | |
| | Free range | 646 | 14 (2.17) | | 30 (2) | 7 (23.33) | |
| Feed | Commercial feed | 698 | 55 (7.88) | | 49 (36) | 27 (55.10) | * |
| | Local feed | 802 | 32 (3.99) | | 28 (2) | 5 (7.14) | |

Note: Out of 1,500 samples collected from 77 farms/flocks, 87 tested positive for coccidiosis, with 32 farms reporting positive cases and 38 farms using commercial feed containing coccidiostats.

Risk factor categories marked with * show statistically significant differences (p<0.05) in prevalence based on Chi-square test.

to the manufacturer instructions with initial breaking of oocyst wall by using a Bead beater. Extracted DNA samples were tested by PCR using the *Eimeria* species specific primers (Table 3, Fig 1).

All primers used in this study were synthesized by Macrogen Inc., Korea, and were MOPC-purified to ensure high specificity and quality for PCR amplification. PCR reactions were performed in a final volume of 25 µL, containing 12.5 µL of 2×PCR Master Mix (Takara, Japan), 0.5 µM of each forward and reverse primer, 2 µL of template DNA, and nuclease-free water to adjust the final volume. Positive controls included DNA extracted from field isolated *Eimeria* species confirmed by sequence analysis. Negative controls consisted of PCR reactions with nuclease-free water in place of template DNA. These controls were included during PCR run to validate the amplification results and rule out contamination.

PCR was performed in miniPCR (Oxford) with the following cyclic conditions: 95°C for 5 min, 30 cycles at 94°C for 30 sec, individual annealing temperature (Table 3) for 30 sec, 72°C for 30 sec, and a final extension of 72°C for 7 min. The amplified product of PCR assay was analysed by gel electrophoresis on a 1.5% agarose gel and stained with ethidium bromide for visualisation of gel under UV illumination.

Species-specific primers targeting the internal transcribed spacer-1 (ITS-1) region were used for initial PCR detection of *Eimeria* spp., as described by Haug et al. [20]. The species of *Eimeria* were confirmed by sequencing the representative PCR products obtained from PCR using species specific and universal primers (Table 3). PCR products were run on an

**Table 3. PCR primers targeting the ITS-1 of the seven valid *Eimeria* species [52].**

| Species | Primer sequence 5′ - 3′ | Expected Amplicon Size (bp) | Annealing temperature (°C) |
|---|---|---|---|
| Eimeria spp. (universal primer) | 5′-AAGTTGCGTAAATAGAGCCCT-3′<br>5′-AGACATCCATTGCTGAAAG-3′ | 400–750 | 56 |
| *E. acervulina* | 5′-GGGCTTGGATGATGTTTGCTG-3′<br>5′-GCAATGATGCTTGCACAGTCAGG-3′′ | 145 | 57 |
| *E. brunetti* | 5′-CTGGGGCTGCAGCGACAGGG-3′<br>5′-ATCGATGGCCCCATCCCGCAT-3′ | 183 | 65 |
| *E. maxima* | 5′-TTGTGGGGCATATTGTTGTGA-3′<br>5′-CWCACCACTCACAATGAGGCAC-3′ | 162 | 55 |
| *E. mitis* | 5′-GTTTATTTCCTGTCGTCGTCTCGC-3′<br>5′-GTATGCAAGAGAGAATCGGGATTCC-3′ | 330 | 57 |
| *E. necatrix* | 5′-AGTATGGGCGTGAGCATGGAG-3′<br>5′-GATCAGTCTCATCATAATTCTCGCG-3′ | 160 | 53 |
| *E. praecox* | 5′-CATCGGAATGGCTTTTTGAAAGCG-3′<br>5′-GCATGCGCTAACAACTCCCCTT-3′ | 215 | 56 |
| *E. tenella* | 5′-AATTTAGTCCATCGCAACCCTTG-3′<br>5′-CGAGCGCTCTGCATACGACA-3′ | 278 | 55 |

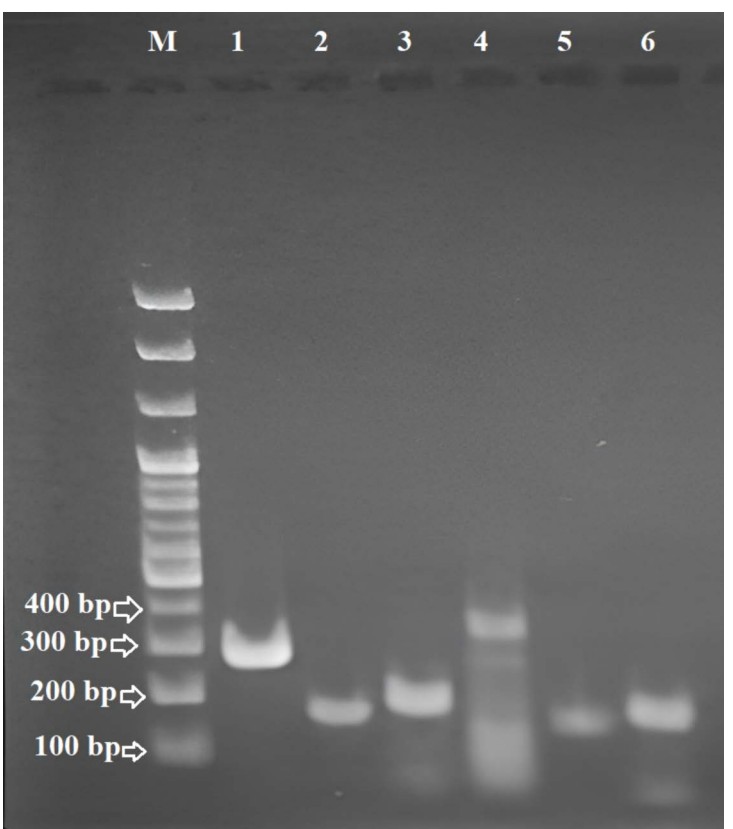

**Fig 1. Agarose gel electrophoresis showing PCR amplification of representative *Eimeria* species from field samples.** Lane M: 100 bp DNA ladder; Lane 1: *E. tenella* (278 bp) from Mymensingh (Sutiakhali); Lane 2: *E. acervulina* (146 bp) from Mymensingh (Vabokhali); Lane 3: *E. brunetti* (183 bp) from Rajshahi (Puba); Lane 4: *E. mitis* (330 bp) from Comilla (Sarail); Lane 5: *E. maxima* (162 bp) from Dhaka (Savar); Lane 6: *E. necatrix* (162 bp) from Rangpur (Gangachara).

agarose gel, cut, purified, and then single-stranded products were generated using cycle sequencing PCR with forward or reverse primers. The purified products were run on a Sanger machine (ABI 3730 xl) using the dideoxy chain termination method at GeneCreate Biotech, China [21].

Representative sequences that were identified in this study were deposited in the GenBank (Accession No. LC855064 -*Eimeria tenella*; LC855065 – *Eimeria necatrix*; LC855066 – *Eimeria brunetti*; LC855067- *Eimeria maxima*; LC855068- *Eimeria mitis* and LC855069- *Eimeria acervulina*)

## Statistical analysis

The data were analyzed using statistical R software. Descriptive statistics were calculated to summarize the prevalence of coccidiosis across different regions and farming systems. The association between potential risk factors (age, sex, breed, rearing system and feed) and the presence of *Eimeria* oocysts was assessed using chi-square tests. A p-value of <0.05 was considered statistically significant.

## Ethical considerations and permits

Fecal samples were collected non-invasively from chicken farm litter or from the ground surface in homestead free-ranging systems, without any handling or disturbance to the chickens. No animals were restrained, touched, or harmed during the sample collection process. Therefore, ethical approval was not required for this study. All samples were collected with prior verbal consent from farm owners or household heads. Since the study involved environmental sampling from private properties and did not include animal experimentation, no specific government permits were necessary. Field access was granted by the respective owners, and no protected areas or government-owned facilities were involved.

## Results

### *Eimeria* infection status in chickens

Of the 1500 faecal samples examined microscopically, *Eimeria* oocysts were detected in 87 (5.8%) samples out of 77 farms/flocks. The prevalence of *Eimeria* infection varied significantly across farms or flocks based on breed and management systems (Table 4). *Eimeria* oocysts were observed in 32 farms/flocks (41.56%) out of 77. Among exotic breeds, Broiler farms had a prevalence rate of 76.92% (10 out of 13 farms), followed by Sonali farms with a 66.67% prevalence (8 out of 12 farms). Fayoumi farms, however, reported no positive case. For native breeds, Aseel farms had the highest prevalence at 100% (5 out of 5 farms), followed by Deshi farms with 38.89% (7 out of 18 farms). Naked Neck and Hilly flocks demonstrated 0% prevalence (Table 2). In our study six species of *Eimeria* have been identified from various regions by PCR, including *E. tenella, E. acervulina, E. brunetti*, *E. mitis E. necatrix,* and *E. maxima.*

**Table 4.  Risk factor analysis of *Eimeria* infection in different breeds of chicken.**

| Risk Factor | Chi Sq value | Category | Odds Ratio | 95% CI | p-value |
|---|---|---|---|---|---|
| Breed | $\chi^2 = 20.76$, df = 2, p < 0.001 | Exotic vs. Native | 2.01 | (1.29- 3.13) | 0.002 |
| | | Mixed vs. Native | 3.16 | (0.93- 10.76) | 0.066 |
| Sex | $\chi^2 = 9.37$, df = 1, p = 0.002 | Male vs. Female | 2.07 | (1.29- 3.32) | 0.002 |
| Age | $\chi^2 = 300.45$, df = 1, p < 0.001 | Young vs. Adult | 200.76 | (49.07-821.39) | <0.001 |
| Rearing System | $\chi^2 = 66.25$, df = 1, p < 0.001 | Intensive with litter vs. Free range | 8.13 | (4.54- 14.56) | <0.001 |
| Feed | $\chi^2 = 10.11$, df = 1, p = 0.001 | Commercial vs. Local | 2.06 | (1.32- 3.20) | 0.001 |

Note: For the breed category, we used Native as the reference group for comparison. An odds ratio of 1 would indicate no association, while values further from 1 (in either direction) indicate stronger associations.

## Breed

The average Oocyst Per Gram (OPG) levels varied significantly across different breeds and management systems, reflecting the intensity of *Eimeria* infections. The prevalence of infection varied significantly across different breeds ($\chi^2 = 20.76$, df = 2, p < 0.001) with 7.96%, 4.13% and 12% infection of exotic, native and mixed population, respectively. The odds of infection in exotic breeds were significantly higher compared to native breeds (OR = 2.01, 95% CI: 1.29–3.13, p = 0.002) (Table 4). Exotic breeds such as Broiler (13.98%) and Sonali (12.09%) showed significantly higher infection rates compared to native breeds like Deshi (3.40%) and Aseel (18.40%) (p < 0.05). Fayoumi, Naked Neck and Hilly showed no cases of infection.

Among exotic breeds, Broilers had the highest average OPG range (33–1967) with 13.98% prevalence of infection, followed by Sonali (133–1066) with 12.09% prevalence, while Fayoumi showed no positive cases. For native breeds, Aseel had the highest average OPG (66–500) with 18.40% prevalence, whereas Deshi chickens exhibited lower OPG levels (33–133) with 3.40% prevalence. The mixed population also displayed varying OPG levels, with Broiler+Deshi combinations reaching 133 at the higher end.

Among *Eimeria* species *E. tenella* was detected in 64 samples (62.07%) and in all breeds with the highest occurrence in Aseel (23 cases) (Table 5). *E. acervulina* was the second most common species (25.28%), found in 22 samples from Deshi, Broiler and Sonali breeds. Other species, including *E. brunetti* (2.3%), *E. mitis* (3.45%), *E. necatrix* (3.45%), and *E. maxima* (3.45%), were less common and showed limited distribution across breeds. *E. tenella*, *E. acervulina*, and *E. necatrix* were detected in infected Deshi native chickens. *E. mitis* was detected in Broiler and Sonali birds while *E. brunetti* was identified exclusively in the Sonali breed. Notably, *E. necatrix* was detected in both Deshi native chickens and Broilers. Mixed infection was identified in two samples with *E. tenella*+ *E. mitis* infection only in Sonali and *E. acervulina*+*E. maxima* in Broiler breeds.

Breed age was found to be the strongest risk factor associated with *Eimeria* infection ($\chi^2 = 300.45$, df = 1, p < 0.001). Young breeds had a substantially higher prevalence of 25.45%, whereas adult chickens had a prevalence of only 0.17% (OR = 200.76, 95% CI: 49.07–821.39, p < 0.001). The wide confidence interval (49.07–821.39) suggests significant variability, likely due to the very low prevalence in adult chickens.

There was a significant association between sex and the prevalence of infection ($\chi^2 = 9.37$, df = 1, p = 0.002). Males had a higher prevalence (9.80%) compared to females (4.98%) with the odds of 2.07 (OR = 2.07, 95% CI: 1.29–3.32, p = 0.002).

## Rearing system

Samples were collected from various chicken types reared under different management systems in Bangladesh. Deshi chickens, typically raised as flocks in free-range systems. Naked Neck chickens were predominantly reared in free-range

**Table 5. PCR detection of *Eimeria* species in microscopically positive samples from different breeds under different management systems.**

| *Eimeria* species | Breed; Positive samples (% infection) | | | |
|---|---|---|---|---|
| Total detection (87) | Deshi (12) | Aseel (23) | Sonali (25) | broiler (27) |
| *E. tenella* (54) | 8 | 23 | 13 | 10 |
| *E. acervullina* (22) | 3 | 0 | 7 | 12 |
| *E. brunetti* (2) | 0 | 0 | 2 | 0 |
| *E. mitis* (2) | 0 | 0 | 1 | 2 |
| *E. necatrix* (3) | 1 | 0 | 0 | 2 |
| *E. maxima* (3) | 0 | 0 | 2 | 1 |
| *E. tenella* + *E. mitis* | 0 | 0 | 1 | 0 |
| *E. acervulina* + *E. maxima* | 0 | 0 | 0 | 1 |

systems, except for two commercial flock kept in a semi-intensive system without litter. Aseel birds were exclusively raised in intensive systems with litter. Hilly chickens were primarily reared in free-range systems, with one flock managed semi-intensively without litter. All Sonali and Broiler chickens were kept in intensive systems with litter, whereas Fayoumi chickens were observed in intensive systems both with and without litter, as well as in semi-intensive systems with litter.

The rearing system was significantly associated with the prevalence of *Eimeria* infection ($\chi^2 = 66.25$, df = 1, p < 0.001). Chickens raised under intensive systems with litter had the a significantly higher prevalence (15.27%) compared to those in litter-free, semi-intensive, or free-range systems (p < 0.05). The odds of coccidiosis were significantly higher in chickens raised in intensive systems with litter compared to those raised in a free-range system (OR = 8.13, 95% CI: 4.54–14.56, p < 0.001). Notably, no oocysts were detected in samples collected from chickens reared in intensive systems without litter, semi-intensive systems with litter, or semi-intensive systems without litter. The highest prevalence of coccidiosis was observed in farms practicing a litter-based intensive system (68%), followed by intensive systems without litter (42.86%). Semi-intensive systems demonstrated prevalence rates of 30% with litter and 40% without litter.

When analyzed based on rearing systems, intensive farms with litter exhibited the highest prevalence at 68% (17 out of 25 farms), followed by Intensive without litter at 42.86% (3 out of 7) and free-range farms at 23.33% (7 out of 30 farms). Farms under semi-intensive systems with and without litter showed 30% and 40% prevalence, respectively.

### Feed

The type of feed was found to have a significant impact on the prevalence of coccidiosis ($\chi^2 = 10.11$, df = 1, p = 0.001). Chickens fed commercial feed had a higher prevalence (7.88%) compared to those fed local feed (3.99%) with the odds of 2.06 (OR = 2.06, 95% CI: 1.32–3.20, p = 0.001). The percentage of farms using coccidiostats varied across breeds and management systems. Among exotic breeds, 100% of Sonali and Broiler farms utilized coccidiostats, highlighting their intensive management practices and higher vulnerability to coccidiosis. Conversely, Fayoumi farms had a lower coccidiostat usage rate (60%). For native breeds, only Aseel farms consistently used coccidiostats (100%), while Deshi and Naked Neck farms had minimal usage rates of 5.56% and 6.67%, respectively (Table 2).

## Discussion

### Infection status of *Eimeria* in chicken

Rahman et al. [1] who reported 5.8% prevalence of coccidiosis in Gazipur and Badruzzaman et al. [22] reported 7.87% coccidiosis in Sylhet regions of Bangladesh which are in accordance with the current study. Our study identified six species of *Eimeria* (*E. tenella, E. acervulina, E. brunetti, E. mitis E. necatrix,* and *E. maxima*) among the samples corrected from different regions of Bangladesh. In a study by Alam et al. [17], five *Eimeria* species, including *E. tenella, E. necatrix, E. acervulina, E. brunetti, and E. mitis* were identified in broiler farms in Bangladesh using the ITS1 region specific marker. In contrast, Siddiki et al. [12] detected seven species, including *E. precox*. Neither this study nor Alam's study detected *E. precox*, which could be attributed to methodological differences, as Siddiki et al. [12] used conventional PCR targeting unique single-copy sequences derived from sequence-characterized amplified region (SCAR) markers. Research in Egypt isolated *E. tenella, E. acervulina, E. necatrix*, and *E. praecox* from native chickens [23] which in accordance with our study expect *E. praecox*. In India, next-generation sequencing revealed the presence of all seven recognized *Eimeria* species in both commercial and indigenous chickens, with *E. tenella* and *E. necatrix* dominating [24]. A study in Pakistan identified four *Eimeria* species in layer chickens, with *E. tenella* being the most prevalent (39.93%), and found higher infection rates in young chickens compared to adults [25].

Among *Eimeria* species *E. tenella* was detected in 62.07% of samples and in all breeds of this study. This finding is consistent with previous studies, such as Alzib et al. [26], which identified *E. tenella* as a highly pathogenic species responsible for severe cecal coccidiosis, particularly in intensively managed poultry systems. In Iran, a study found 64% of native chickens infected, with *E. tenella* being the most prevalent species [27]. Broilers showed the highest

diversity of *Eimeria* species, including *E. acervulina* (44.45%) and less common species like *E. mitis* and *E. necatrix* (7.41% each), indicating their susceptibility to multiple infections likely due to their intensive farming conditions [28]. Sonali chickens exhibited moderate prevalence rates, with infections of *E. brunetti* and *E. maxima* (8.33% each), underscoring their vulnerability under semi-intensive systems. Indigenous chicken varieties in Bangladesh, such as Common Deshi, Hilly, and Naked Neck, are known for their disease resistance and survivability [29]. In this study no or lower overall infection rates of native chicken, suggesting inherent resistance possibly linked to genetic and environmental factors [30]. None of the sample was positive for *E. praecox* which is in accordance with findings of Bhaskaran et al. [31] from India.

## Breed

Studies have shown that exotic breeds, including Sonali and Broilers, tend to have higher prevalence rates compared to native breeds (Table 2). In Nigeria, exotic breeds exhibited a 42.4% prevalence rate, while in Bangladesh, coccidiosis was found to be the most common disease in Sonali chickens, with a prevalence of 21.27% [32,33]. Another study in Bangladesh reported coccidiosis as the most frequent disease in Sonali chickens, affecting 49.2% of cases [9]. These higher rates in exotic breeds may be attributed to poor management practices and biosecurity measures. Genetic predispositions and the management practices associated with exotic breeds often exacerbate their vulnerability. In contrast, native breeds like Deshi have shown lower susceptibility to various diseases, including coccidiosis [34]. In addition, the low prevalence is associated with extensive farming conditions, with gradual exposure to low levels of oocysts, and the development of immunity [35,36]. The relatively lower prevalence in Deshi chickens may indicate the potential for using indigenous breeds in coccidiosis management programs.

The study observed higher prevalence (9.80%) of male birds compared to females (4.98%). This result is consistent with the previous studies who reported a higher prevalence of poultry coccidiosis in male than female chickens [37]. Similarly, Wondimu et al. [38] reported higher prevalence of coccidiosis in male (43.6%) than female chicken (41.2%). However, most studies found no significant difference in prevalence between male and female chickens [39–41], although one study reported higher rates in females [42]

For coccidiosis, age is consistently identified as a major risk factor, with young chickens showing higher infection rates compared to adults [32,43,44]. Lower prevalence in older birds due to the immunity they develop from early exposure to the infection.

## Rearing system

Chickens raised under intensive systems with litter had the highest prevalence (15.27%), while those reared in a free-range system had a much lower prevalence (2.17%). These results are consistent with findings from Negash et al. [43], Lawal et al. [32] and Adem et al. [45] where intensive systems were associated with poor litter management and overcrowding, providing favorable conditions for oocyst sporulation.

In contrast to commercial farms, free-range systems demonstrated a significantly lower prevalence of coccidiosis (7.14%). This is likely due to reduced bird density, natural feed sources, and minimal use of antibiotics or anticoccidials [46], which may help maintain healthier gut microbiota and stronger immunity in these chickens due to more natural rearing conditions [32]. In addition, the scavenging village chickens are also less likely to ingest pathogenic level of the coccidian oocysts during feeding [35,36].

Coccidiosis does not occur typically in all chickens in a flock simultaneously despite the presence of oocysts [47], as resistance and susceptibility vary among individuals [48]. Factors influencing the spread include population density, environmental conditions affecting oocyst development, and the development of immunity in previously infected birds [48]. However, the results underscore the higher risk of coccidiosis in farms with intensive management, exotic breeds, and mixed populations compared to native breeds and extensive systems.

### Feed

Commercial poultry farms often rely on anticoccidial feed containing ionophores, non-ionophore chemicals, antibiotics, and growth promoters as part of their shuttle programs. Similarly, anticoccidial shuttle programs in Bangladesh are reliance on combinations of maduramycin, nicarbazin, salinomycin, and others. Among the coccidiostat used farms (38 farms), oocysts were detected in 32 farms (84.21%). Farms using commercial feed exhibited a higher prevalence of coccidiosis (55.10%) compared to free-range systems using local feed, which showed only 7.14% prevalence. Rony et al. [5] reported similar findings of higher coccidiosis prevalence in commercial systems of Bangladesh.

Coccidiostat usage was notably higher in intensive systems, particularly in those with litter, where 80% of farms included it in their practice which is in accordance with the finding of Kadykalo et al. [49]. Semi-intensive systems, both with and without litter, also had low coccidiostat usage rates (8% and 3%, respectively). This trend indicates that farms with intensive management and commercial feed are more dependent on coccidiostats, likely due to higher disease risks and production demands. Despite the use of coccidiostats, infections still occur with low OPG, because most of anticoccidial drugs just reduce the clinical symptoms but cannot inhibit the infections itself completely [50]. Therefore, the used anticoccidial drugs effectively reduced the OPGs and inhibited to develop the clinical symptoms in the present study.

Commercial feeds may contain subtherapeutic levels of anticoccidial drugs, which can lead to lower effectiveness against coccidiosis [51]. Countries with stringent biosecurity measures, improved management practices, and vaccination programs tend to report lower prevalence rates. The lack of such measures in Bangladesh, coupled with management practices, likely explains the higher prevalence in commercial systems.

Local feed used in free-range systems is associated with a very low prevalence in this study (Table 2). Local feed often contains natural ingredients that may improve gut health and immunity [52].

The overall lower rate of *Eimeria* infection reported in this study could reflect the current status of coccidiosis in Bangladesh, largely influenced by feeding and rearing practices in commercial poultry production. One critical factor contributing to this trend is the widespread use of feed supplemented with anticoccidial drugs. These medications are routinely administered as a preventive measure to control *Eimeria* infections, significantly reducing the prevalence and severity of coccidiosis in many broiler farms of Bangladesh.

### Limitation of the study

This study has several limitations that should be acknowledged. The cross-sectional design captures prevalence at a single point in time, making it challenging to determine seasonal variations or long-term trends in *Eimeria* spp. infections. Additionally, reliance on fecal sampling, while useful for detecting oocysts, does not directly assess clinical severity or subclinical infections, potentially underestimating the true impact of coccidiosis. The molecular analysis, though effective for species identification, did not investigate genetic variations or emerging strains that could influence disease dynamics. Furthermore, data on management and feeding practices were collected through farmer surveys, which may be subject to recall bias or inaccuracies regarding feed formulations, anticoccidial usage, and disease prevention measures. Another important consideration is that native chickens reared in free-range systems consume a highly variable diet, which was not systematically analyzed for its potential influence on coccidiosis prevalence. Future studies should address these limitations by incorporating longitudinal sampling, histopathological analysis, genetic diversity assessments, and controlled dietary evaluations to provide a more comprehensive understanding of *Eimeria* infections in different poultry farming systems.

### Conclusion

This study provides a comprehensive assessment of *Eimeria* infection prevalence in different chicken breeds and management systems across Bangladesh for the first time. The overall prevalence of *Eimeria* infection was 5.8%, with

significant variations across breeds, age groups, rearing systems, and feeding practices. Among exotic breeds, Broilers and Sonali chickens exhibited the highest infection rates, whereas Fayoumi chickens showed no infections. Among native breeds, Aseel chickens had the highest prevalence, while Naked Neck and Hilly chickens remained uninfected. The predominant *Eimeria* species identified were *E. tenella* and *E. acervulina*, with *E. brunetti*, *E. mitis*, *E. necatrix*, and *E. maxima* being less common.

The study highlights that younger birds are significantly more susceptible to infection, with males exhibiting a higher prevalence than females. Management systems played a crucial role, with intensive rearing on litter being the most significant risk factor for coccidiosis, while free-range systems demonstrated the lowest prevalence. Additionally, chickens fed commercial diets had a higher risk of infection compared to those consuming locally sourced feed.

These findings emphasize the need for targeted control measures, particularly in high-risk breeds and intensive farming systems. Improved biosecurity, optimized litter management, and strategic coccidiostat use could help mitigate *Eimeria* infections. Future research should focus on resistance mechanisms in naturally resistant breeds and explore alternative, sustainable approaches to coccidiosis prevention in poultry production.

## Supporting information

**S1_raw_images. Agarose gel electrophoresis showing PCR amplification of representative *Eimeria* species from field samples.**
(PDF)

**S2_raw_images. PCR Detection of *Eimeria* spp. from Field Samples.**
(PDF)

## Author contributions

**Conceptualization:** Md. Shahiduzzaman.

**Data curation:** Md. Afazur Rahman, Kazi Farah Tasfia, Makoto Matsubayashi.

**Formal analysis:** Md. Afazur Rahman, A R M Beni Amin, Kazi Farah Tasfia, Makoto Matsubayashi.

**Funding acquisition:** Md. Shahiduzzaman.

**Investigation:** Md. Shahiduzzaman, Kazi Farah Tasfia.

**Methodology:** Md. Shahiduzzaman, Md. Afazur Rahman, A R M Beni Amin, Kazi Farah Tasfia.

**Project administration:** Md. Shahiduzzaman.

**Supervision:** Md. Shahiduzzaman.

**Validation:** A R M Beni Amin, Kazi Farah Tasfia.

**Visualization:** Makoto Matsubayashi.

**Writing – original draft:** Md. Shahiduzzaman.

**Writing – review & editing:** Md. Shahiduzzaman, Makoto Matsubayashi.

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
