## [Decision Letter · Decision Letter 0]

Dear Dr. Shahiduzzaman,

Thank you for submitting your manuscript to PLOS ONE. After careful consideration, we feel that it has merit but does not fully meet PLOS ONE’s publication criteria as it currently stands. Therefore, we invite you to submit a revised version of the manuscript that addresses the points raised during the review process.

**ACADEMIC EDITOR:** The manuscript needs huge effort to improve in consedration to the reviewers' comments specially the statistics.** **

We look forward to receiving your revised manuscript.

Kind regards,

Shawky M Aboelhadid, PhD

Academic Editor

PLOS ONE

Journal Requirements:

“BAS-USDA Program/2023/26(12)”

“This research was funded by the Bangladesh Academy of Science (BAS)-USDA Endowment Program (BAS-USDA Program/2023/26(12)). The authors extend their gratitude to Prof. Dr. Tofazzal Islam, BAS Fellow, for his valuable monitoring of the project.”

“BAS-USDA Program/2023/26(12)”

6. We note that Figure 1 in your submission contain [map/satellite] images which may be copyrighted. All PLOS content is published under the Creative Commons Attribution License (CC BY 4.0), which means that the manuscript, images, and Supporting Information files will be freely available online, and any third party is permitted to access, download, copy, distribute, and use these materials in any way, even commercially, with proper attribution. For these reasons, we cannot publish previously copyrighted maps or satellite images created using proprietary data, such as Google software (Google Maps, Street View, and Earth). For more information, see our copyright guidelines: http://journals.plos.org/plosone/s/licenses-and-copyright.

 A. You may seek permission from the original copyright holder of Figure 1 to publish the content specifically under the CC BY 4.0 license. 

B. If you are unable to obtain permission from the original copyright holder to publish these figures under the CC BY 4.0 license or if the copyright holder’s requirements are incompatible with the CC BY 4.0 license, please either i) remove the figure or ii) supply a replacement figure that complies with the CC BY 4.0 license. Please check copyright information on all replacement figures and update the figure caption with source information. If applicable, please specify in the figure caption text when a figure is similar but not identical to the original image and is therefore for illustrative purposes only.

7. PLOS ONE now requires that authors provide the original uncropped and unadjusted images underlying all blot or gel results reported in a submission’s figures or Supporting Information files. This policy and the journal’s other requirements for blot/gel reporting and figure preparation are described in detail at https://journals.plos.org/plosone/s/figures#loc-blot-and-gel-reporting-requirements and https://journals.plos.org/plosone/s/figures#loc-preparing-figures-from-image-files. When you submit your revised manuscript, please ensure that your figures adhere fully to these guidelines and provide the original underlying images for all blot or gel data reported in your submission. See the following link for instructions on providing the original image data: https://journals.plos.org/plosone/s/figures#loc-original-images-for-blots-and-gels.  

Reviewers' comments:

Reviewer's Responses to Questions

**Comments to the Author**

1. Is the manuscript technically sound, and do the data support the conclusions?

Reviewer #1: Partly

Reviewer #2: Yes

2. Has the statistical analysis been performed appropriately and rigorously?

Reviewer #1: Yes

Reviewer #2: Yes

3. Have the authors made all data underlying the findings in their manuscript fully available?

Reviewer #1: Yes

Reviewer #2: Yes

4. Is the manuscript presented in an intelligible fashion and written in standard English?

Reviewer #1: Yes

Reviewer #2: Yes

Reviewer #1: The study ‘Molecular Detection and Risk Factors of Eimeria in Native and Exotic Chickens under Varying Management Systems in Bangladesh’ was conducted by Dr. Rahman and its groups, and reported a variety of interesting results and providing valuable information to relevant poultry coccidiosis fields.

I have read the paper wholeheartedly, and here are some questions:

L58-60/Figure 1: You mentioned in the introduction that Bangladesh's hot and humid climate provides favorable conditions for the development of Eimeria, but your measured risk factors do not provide any temperature, humidity, or rainfall data.

Why didn't you measure the key elements of Eimeria development mentioned in the introduction and use them as risk factors?

L415-417: The question I want to ask the most is what the meaning of this comprehensive study is, which is already written in the limitations of the study.

Even if various positive samples were found in different regions of Bangladesh, unless data on the phenotype represented by the chickens, e.g., lesion score, are submitted, this study feels inconclusive.

In addition, as you have already mentioned, no relevant data have been submitted on whether oocysts actually exhibit clinically pronounced infection symptoms, even if they are found.

Therefore, I think this study you conducted is insufficient to draw any conclusions despite the time and effort it took.

L639: In this table format, I cannot understand which one has more risk factors than the others.

It can be seen that there was a significant difference between the two categories.

It would be good to choose a different statistical method or use a method that can be indicated.

L648: A positive sample itself cannot mean serious clinical symptoms.

Beyond just a sample showing positive or negative, is there an experimentally infected control group?

For example, have you confirmed that similar band sizes and locations occur in chickens that have been orally administered Eimeria enough to cause clinical symptoms?

Comments:

Table 2: Typo: Brioler > broiler

Table 3: Typo in primer sequence 'CWCACCACTCACAATGAGGCAC'

Table 5: Typo: Brioler > broiler

Reviewer #2: This cross-sectional study provides valuable epidemiological data on Eimeria spp. prevalence in Bangladeshi chickens, combining field surveys with molecular characterization. The large sample size (n=1,500) and multi-district approach strengthen the findings. The identification of commercial feed and intensive systems as risk factors offers practical implications for poultry management. However, methodological details require clarification to ensure reproducibility, and some analytical aspects need expansion. The work would benefit from improved technical descriptions and better integration between methodology and results.

1. Sample Collection Methodology:

The fecal sample collection protocol lacks critical details. Please specify:Number of samples collected per farm/flock.

Minimum fecal quantity per sample (grams). Clarify whether pooled samples were used or individual bird samples were maintained separately.

2. Oocyst Detection Protocol (Major):

The salt flotation method requires validation details. State whether McMaster chamber counts followed standard OPG calculations (oocysts per gram). Cite original references for both flotation and McMaster techniques

3. Molecular Methods (Major):

PCR protocols need technical enhancement. Specify synthesis source (company) and purification grade. State primer concentrations in master mixes. Include positive/negative controls used. Clarify species differentiation method: Were species-specific primers used, or was sequencing performed?

4. Figure 2 Concerns (Major): The gel electrophoresis image requires, Clear lane labels indicating sample types (test samples vs controls) . Provide representative gel images of clinical samples (not just positive controls) in the main text or supplementary materials.

5. Resolve discrepancy between methods mentioning phylogenetic analysis and missing results, Either present the evolutionary tree with bootstrap values.

**Do you want your identity to be public for this peer review?** For information about this choice, including consent withdrawal, please see our Privacy Policy

Reviewer #1: No

Reviewer #2: No

---

## [Author Response · Author response to Decision Letter 1]

30 May 2025

Journal Requirements (Academic Editor):

Author Response:

Thank you for the reminder. We have thoroughly reviewed and revised our manuscript to ensure full compliance with PLOS ONE's style requirements, including formatting, figure legend presentation, and file naming conventions.

Author Response:

Ethical considerations and permits (The information were added in Methodology section of revised manuscript) Lines 219-238.

Fecal samples were collected non-invasively from chicken farm litter or from the ground surface in homestead free-ranging systems, without any handling or disturbance to the chickens. No animals were restrained, touched, or harmed during the sample collection process. Therefore, ethical approval was not required for this study. All samples were collected with prior verbal consent from farm owners or household heads. Since the study involved environmental sampling from private properties and did not include animal experimentation, no specific government permits were necessary. Field access was granted by the respective owners, and no protected areas or government-owned facilities were involved.

Author Response:

Thank you for your comment regarding the grant information. I did not find a separate "Financial Disclosure" section in the submission system. However, I located the "Funding Information" section, where I have correctly provided the grant number for the award: BAS-USDA Program/2023/26(12). Please let me know if any further clarification or revision is needed.

“BAS-USDA Program/2023/26(12)”

Author Response:

The funders had no role in study design, data collection and analysis, decision to publish, or preparation of the manuscript

“This research was funded by the Bangladesh Academy of Science (BAS)-USDA Endowment Program (BAS-USDA Program/2023/26(12)). The authors extend their gratitude to Prof. Dr. Tofazzal Islam, BAS Fellow, for his valuable monitoring of the project.”

“BAS-USDA Program/2023/26(12)”

Author Response:

We removed the funding-related text from the manuscript (Acknowledgments section). The funding statement will read as “BAS-USDA Program/2023/26(12)”.

6. We note that Figure 1 in your submission contain [map/satellite] images which may be copyrighted. All PLOS content is published under the Creative Commons Attribution License (CC BY 4.0), which means that the manuscript, images, and Supporting Information files will be freely available online, and any third party is permitted to access, download, copy, distribute, and use these materials in any way, even commercially, with proper attribution. For these reasons, we cannot publish previously copyrighted maps or satellite images created using proprietary data, such as Google software (Google Maps, Street View, and Earth). For more information, see our copyright guidelines: http://journals.plos.org/plosone/s/licenses-and-copyright.

Author Response:

We have removed the figure from submission and deleted the Figure caption and citation in the texts.

7. PLOS ONE now requires that authors provide the original uncropped and unadjusted images underlying all blot or gel results reported in a submission’s figures or Supporting Information files. This policy and the journal’s other requirements for blot/gel reporting and figure preparation are described in detail at https://journals.plos.org/plosone/s/figures#loc-blot-and-gel-reporting-requirements and https://journals.plos.org/plosone/s/figures#loc-preparing-figures-from-image-files. When you submit your revised manuscript, please ensure that your figures adhere fully to these guidelines and provide the original underlying images for all blot or gel data reported in your submission. See the following link for instructions on providing the original image data: https://journals.plos.org/plosone/s/figures#loc-original-images-for-blots-and-gels.

Author Response:

The original uncropped and unadjusted images are uploaded as a Supporting Information file 'S1_raw_images' (PDF) and 'S2_raw_images'

Response to Reviewers

Reviewer #1: The study ‘Molecular Detection and Risk Factors of Eimeria in Native and Exotic Chickens under Varying Management Systems in Bangladesh’ was conducted by Dr. Rahman and its groups, and reported a variety of interesting results and providing valuable information to relevant poultry coccidiosis fields.

I have read the paper wholeheartedly, and here are some questions:

Reviewer Comment:

L58-60/Figure 1: You mentioned in the introduction that Bangladesh's hot and humid climate provides favorable conditions for the development of Eimeria, but your measured risk factors do not provide any temperature, humidity, or rainfall data.

Why didn't you measure the key elements of Eimeria development mentioned in the introduction and use them as risk factors?

Author Response:

Thank you for this thoughtful observation. We fully acknowledge that climatic variables such as temperature, humidity, and rainfall are critical to the sporulation and survival of Eimeria oocysts and are thus key factors influencing the epidemiology of coccidiosis. Our mention of Bangladesh’s hot and humid climate in the introduction was intended to provide contextual background, as these general environmental conditions are known to create a conducive setting for the parasite’s development.

However, the current study was designed as a cross-sectional field survey focusing on breed susceptibility, management practices, feeding systems, and rearing conditions as primary risk factors. Given the resource and logistical constraints, real-time and location-specific meteorological data were not collected during sampling. Moreover, since sampling occurred over diverse regions and time frames, integrating consistent and reliable climatic data across all sites would have required a separate longitudinal study design and collaboration with national meteorological services.

Instead, we used proxy indicators, such as litter management, housing type, and production system (intensive vs. free-range), which are closely linked to micro-environmental conditions within poultry facilities and have direct relevance to Eimeria transmission dynamics.

We appreciate your suggestion and agree that integrating climatic variables would enhance our understanding of Eimeria epidemiology. We plan to consider this in future longitudinal studies with geospatial mapping and climatic overlays to better capture environmental influences on infection risk.

Reviewer Comment:

L415-417: The question I want to ask the most is what the meaning of this comprehensive study is, which is already written in the limitations of the study.

Even if various positive samples were found in different regions of Bangladesh, unless data on the phenotype represented by the chickens, e.g., lesion score, are submitted, this study feels inconclusive.

In addition, as you have already mentioned, no relevant data have been submitted on whether oocysts actually exhibit clinically pronounced infection symptoms, even if they are found.

Therefore, I think this study you conducted is insufficient to draw any conclusions despite the time and effort it took.

Author Response:

We sincerely thank the reviewer for this critical and insightful comment. We agree that clinical phenotyping, such as lesion scoring and the documentation of symptomatic infections, would significantly strengthen the implications of Eimeria detection and contribute to understanding its true pathological impact.

However, the primary objective of this study was to provide the first comprehensive baseline data on the prevalence and species diversity of Eimeria spp. across a wide geographical range and among different chicken breeds and management systems in Bangladesh using both parasitological and molecular methods. Prior to this study, data on Eimeria infection status especially among native chickens were extremely limited or nonexistent in many regions. By detecting oocysts and identifying six Eimeria species (including E. tenella and E. acervulina as dominant), our findings help define the distribution and breed susceptibility patterns under existing field conditions.

We acknowledge that the absence of lesion scoring, clinical symptom tracking, and histopathological assessments limits conclusions about clinical severity or economic impact, and we clearly stated this in the Limitations section. Still, the identification of oocyst-positive samples across regions and rearing systems is an essential first step to inform future targeted studies that can integrate phenotypic and pathological assessments.

This foundational survey also provides crucial epidemiological context for native vs. exotic breed vulnerability, risk factor associations (e.g., feed type and production system), and the molecular diversity of Eimeria in Bangladesh. These results are already being used to guide ongoing longitudinal and intervention-based research, including work incorporating clinical assessments and anticoccidial resistance monitoring.

While we agree that further work is needed to build on these findings, we believe this study offers meaningful initial insights into Eimeria epidemiology in Bangladesh and establishes a baseline for future, more clinically detailed investigations.

Reviewer Comment:

L639: In this table format, I cannot understand which one has more risk factors than the others.

It can be seen that there was a significant difference between the two categories.

It would be good to choose a different statistical method or use a method that can be indicated.

Author Response:

We thank the reviewer for this valuable comment. We agree that while Table 2 presents detailed descriptive data on prevalence across multiple risk factor categories (e.g., breed, age, sex, feed type, and rearing system), it lacks a statistical summary that clearly communicates the relative strength and significance of these risk factors.

In response to this suggestion, we have taken the following actions to improve clarity and interpretation:

We have now conducted a chi-square test (χ²) for categorical variables (e.g., breed group, rearing system, feed type, sex, and age category) to determine the statistical significance of associations between these risk factors and Eimeria prevalence. Significant differences (p < 0.05) are now clearly indicated in the revised table and in the main text (Lines 262-264; 301-302).

To improve readability, we have redesigned Table 2 to clearly distinguish between statistically significant and non-significant categories. Highlight high-risk categories in bold or with asterisks (*p < 0.05).

We appreciate the reviewer’s feedback, which helped us improve the analytical robustness and clarity of the manuscript.

Reviewer Comment:

L648: A positive sample itself cannot mean serious clinical symptoms.

Beyond just a sample showing positive or negative, is there an experimentally infected control group?

For example, have you confirmed that similar band sizes and locations occur in chickens that have been orally administered Eimeria enough to cause clinical symptoms?

Author Response:

We sincerely thank the reviewer for this insightful comment. We fully agree that a positive PCR result or the presence of Eimeria oocysts in fecal samples does not necessarily indicate clinically significant coccidiosis, particularly in the absence of lesion scoring, clinical signs, or production losses.

As stated in the limitations of our study, this was a cross-sectional, field-based investigation focusing on the natural occurrence and molecular identification of Eimeria spp. under different management systems in Bangladesh. Because of the non-invasive nature of our sampling approach—collection of naturally shed feces from the environment—we did not include an experimentally infected control group or conduct clinical evaluations such as lesion scoring or growth performance monitoring.

We acknowledge that experimental infection studies, where birds are orally challenged with known doses of Eimeria spp. and monitored for clinical symptoms and lesion development, are essential to establish a direct link between molecular findings and pathogenicity. However, our study was designed to characterize field prevalence and species distribution rather than assess disease severity under controlled conditions.

To ensure species-specific identification, we used validated primers, and positive amplicons were gel-purified and sequenced. The identity of the Eimeria species was confirmed via sequence analysis, with representative sequences submitted to GenBank (accession numbers provided in the manuscript).

We appreciate this valuable suggestion and plan to address this important aspect in future studies through controlled infection experiments and longitudinal field trials that incorporate clinical assessments and lesion scoring to correlate molecular findings with pathogenic outcomes.

Comments:

Table 2: Typo: Brioler > broiler

Response: Done

Table 3: Typo in primer sequence 'CWCACCACTCACAATGAGGCAC'

Response: Thank you for pointing this out. We have carefully rechecked the primer sequence ‘CWCACCACTCACAATGAGGCAC’ and confirm that it is correct, as originally described by Haug et al. (2007). These primers are specific for the short variant of the ITS-1 region (GenBank accession no. AF065095), as cited in our manuscript.

Table 5: Typo: Brioler > broiler

Response: Done

Reviewer #2: This cross-sectional study provides valuable epidemiological data on Eimeria spp. prevalence in Bangla

---

## [Decision Letter · Decision Letter 1]

Molecular Detection and Risk Factors of Eimeria in Native and Exotic Chickens under Varying Management Systems in Bangladesh

PONE-D-25-19066R1

Dear Dr. Shahiduzzaman,

We’re pleased to inform you that your manuscript has been judged scientifically suitable for publication and will be formally accepted for publication once it meets all outstanding technical requirements.

Kind regards,

Shawky M Aboelhadid, PhD

Academic Editor

PLOS ONE

Additional Editor Comments (optional):

Reviewers' comments:

Reviewer's Responses to Questions

**Comments to the Author**

Reviewer #1: All comments have been addressed

Reviewer #2: (No Response)

2. Is the manuscript technically sound, and do the data support the conclusions?

Reviewer #1: Yes

Reviewer #2: (No Response)

3. Has the statistical analysis been performed appropriately and rigorously?

Reviewer #1: Yes

Reviewer #2: (No Response)

4. Have the authors made all data underlying the findings in their manuscript fully available?

Reviewer #1: Yes

Reviewer #2: (No Response)

5. Is the manuscript presented in an intelligible fashion and written in standard English?

Reviewer #1: Yes

Reviewer #2: (No Response)

Reviewer #1: The author answered the reviewer's questions appropriately and revised the contents of all the requests for correction.

Reviewer #2: Throughout the manuscript, please ensure the statistical term "p-value" is formatted correctly with an italicized "p"

**Do you want your identity to be public for this peer review?** For information about this choice, including consent withdrawal, please see our Privacy Policy

Reviewer #1: No

Reviewer #2: No

---

## [Editor Report · Acceptance letter]

PONE-D-25-19066R1

PLOS ONE

Dear Dr. Shahiduzzaman,

I'm pleased to inform you that your manuscript has been deemed suitable for publication in PLOS ONE. Congratulations! Your manuscript is now being handed over to our production team.

Kind regards,

on behalf of

Professor Shawky M Aboelhadid

Academic Editor

PLOS ONE